EMBO
*reports*

# Mitochondrial respiration is required to provide amino acids during fermentative proliferation of fission yeast

Michal Malecki[1,2,*] iD, Stephan Kamrad[2,3] iD, Markus Ralser[3] & Jürg Bähler[2,**] iD

## Abstract

**When glucose is available, many organisms repress mitochondrial respiration in favour of aerobic glycolysis, or fermentation in yeast, that suffices for ATP production. Fission yeast cells, however, rely partially on respiration for rapid proliferation under fermentative conditions. Here, we determined the limiting factors that require respiratory function during fermentation. When inhibiting the electron transport chain, supplementation with arginine was necessary and sufficient to restore rapid proliferation. Accordingly, a systematic screen for mutants growing poorly without arginine identified mutants defective in mitochondrial oxidative metabolism. Genetic or pharmacological inhibition of respiration triggered a drop in intracellular levels of arginine and amino acids derived from the Krebs cycle metabolite alpha-ketoglutarate: glutamine, lysine and glutamic acid. Conversion of arginine into these amino acids was required for rapid proliferation when blocking the respiratory chain. The respiratory block triggered an immediate gene expression response diagnostic of TOR inhibition, which was muted by arginine supplementation or without the AMPK-activating kinase Ssp1. The TOR-controlled proteins featured biased composition of amino acids reflecting their shortage after respiratory inhibition. We conclude that respiration supports rapid proliferation in fermenting fission yeast cells by boosting the supply of Krebs cycle-derived amino acids.**

**Keywords** arginine; cellular metabolism; fermentation; respiration; *S. pombe*
**Subject Categories** Metabolism; Signal Transduction

## Introduction

Glucose oxidation in eukaryotic microbes and mammals depends on three core metabolic pathways, glycolysis, the pentose phosphate pathway and the Krebs cycle. These catabolic processes yield energy in the form of ATP, reducing equivalents in the form of NAD(P)H and FADH, as well as a series of intermediates required for anabolic (biosynthetic) metabolism. The flux distributions between all three pathways are dynamically adjusted by the cell to achieve an optimal level of growth and resilience given a specific lifestyle, stress condition or ecological niche (Ralser *et al*, 2007; Peralta *et al*, 2015; Jouhten *et al*, 2016; Nidelet *et al*, 2016; Olin-Sandoval *et al*, 2019). These metabolic adaptations are of fundamental importance for cellular physiology as they determine their tolerance for changing conditions and stress (Lahtvee *et al*, 2016). Moreover, metabolic flux distributions play a fundamental role in biotechnology. Research has put particular emphasis to explain the balance between respiration and fermentation, i.e. the amount of glucose oxidation achieved via anaerobic metabolism (glycolysis and pentose phosphate pathways without oxidative phosphorylation) versus oxidative metabolism, in which all three pathways are active and feed electrons into the oxygen-consuming respiratory chain (Molenaar *et al*, 2009).

Particularly in eukaryotes, rapidly proliferating cells often use anaerobic metabolism even in the presence of oxygen, called aerobic glycolysis, albeit only full glucose oxidation through the respiratory chain maxes out the redox potential stored in glucose and yields the maximum number of ATP molecules. Growth by aerobic glycolysis is often referred to as Warburg effect in cancer cells (Warburg, 1956; Lunt & Vander Heiden, 2011). Related to aerobic glycolysis is the Crabtree effect in yeast cells, reflecting an active suppression of carbohydrate oxidation in the respiratory chain as long as glucose is available (DeRisi *et al*, 1997; Turcotte *et al*, 2010). Conversely, differentiated mammalian cells or yeast cells grown in nutrient-poor media shift their energy metabolism towards respiration (Lunt & Vander Heiden, 2011). Thus, the levels of respiration and fermentation are rebalanced in response to physiological and environmental conditions. ATP need alone is insufficient to predict these metabolic programmes. Better predictability of metabolic states has been achieved with resource allocation models (Basan, 2018), accounting for the cost of synthesising the enzymes (Wortel *et al*, 2018), changes in the surface-to-volume ratio (Szenk *et al*, 2017) and thermodynamic constraints such as metabolic

1  Institute of Genetics and Biotechnology, Faculty of Biology, University of Warsaw, Warsaw, Poland
2  Institute of Healthy Ageing and Research Department of Genetics, Evolution & Environment, University College London, London, UK
3  Molecular Biology of Metabolism Laboratory, The Francis Crick Institute, London, UK
    *Corresponding author. Tel: +48 225923233; E-mail: m.malecki@ucl.ac.uk
    **Corresponding author. Tel: +44 (0) 2031081602; E-mail: j.bahler@ucl.ac.uk

energy dissipation (Niebel *et al*, 2019). These models arrive at the common conclusion that glycolysis is the more efficient pathway and that additional glucose oxidation through the respiratory chain might provide only additional benefit when carbon availability is limited, competition is high, or under conditions where the excretion of small sugars has a negative impact, i.e. when cells require ATP without the need to grow.

Besides providing electron donors for generating ATP, the mitochondrial Krebs cycle is crucial for the synthesis of amino acids and other small biomolecules needed for new cell mass (Hosios *et al*, 2016). Some respiration is required to ensure Krebs cycle activity for the efficient proliferation of human cancer cells growing by aerobic glycolysis (Zhang *et al*, 2014). Upon blocking the respiratory chain by deletion of the mitochondrial genome, cultured mammalian cells need supplementation of uridine and pyruvate for proliferation (King & Attardi, 1989). Uridine synthesis depends on a functional electron transport chain (Grégoire *et al*, 1984), and pyruvate can rescue defects in respiration-mediated aspartate synthesis (Birsoy *et al*, 2015; Sullivan *et al*, 2015). Thus, metabolic products from the Krebs cycle can become limiting for mammalian cell proliferation.

Some single-celled organisms can successfully proliferate in the absence of oxidative phosphorylation. Budding yeast (*Saccharomyces cerevisiae*) is a "petite-positive" yeast, which can grow in anaerobic conditions in the absence of respiration (Chen & Clark-Walker, 1999; Day, 2013). Like uridine and pyruvate supplemented mammalian cells, these yeasts can grow even without a mitochondrial genome. *S. cerevisiae* has evolved adaptations that loosen its dependence on oxidative phosphorylation. An example is the cytoplasmic location of dihydroorotate dehydrogenase, an enzyme that uncouples uridine synthesis from mitochondrial respiration (Nagy *et al*, 1992; Gojković *et al*, 2004). Other examples are the synthesis of glutamic acid and aspartate which normally requires the Krebs cycle metabolites alpha-ketoglutarate and oxaloacetate, respectively. When growing under fermentative conditions, the carbon flux in the Krebs cycle splits to provide alpha-ketoglutarate in the oxidative branch and fumarate in the reductive branch, while the flux between alpha-ketoglutarate and oxaloacetate is inhibited (Gombert *et al*, 2001; Camarasa *et al*, 2003). This strategy allows elimination of succinate dehydrogenase activity and helps to maintain redox homeostasis when the electron transport chain is inhibited (Rodrigues *et al*, 2006). Moreover, when respiration is blocked, *S. cerevisiae* uses the retrograde pathway to activate the glyoxylate cycle that helps to replenish Krebs cycle metabolites (Liao & Butow, 1993; Liu & Butow, 1999).

However, such adaptations are the exception: most yeast species, including fission yeast (*Schizosaccharomyces pombe*), are petite-negative and do not cope well under anaerobic conditions (Heslot *et al*, 1970; Chiron *et al*, 2007). The *S. pombe* laboratory strain is part of a group of natural isolates that show higher respiration during fermentative growth, which is caused by a naturally occurring mutation in pyruvate kinase that limits glycolytic flux (Kamrad *et al*, 2020). We have also reported that respiration is required for rapid cell proliferation during fermentation (Malecki *et al*, 2016). Here we define the respiratory functions in fission yeast that support rapid proliferation. We show that respiration during fermentative growth in *S. pombe* is required for the synthesis of arginine and amino acids derived from alpha-ketoglutarate. The synthesis of these amino acids becomes limiting for cell proliferation if the

respiratory chain is blocked. We further show that blocking respiration under these conditions leads to the transient inhibition of target of rapamycin (TOR) and longer-term amino-acid auxotrophy. Respiration in *S. pombe* cells growing with glucose hence compensates for insufficient amino-acid metabolism.

# Results and Discussion

### Arginine supplementation is necessary and sufficient for rapid cell growth without respiration

We have shown that antimycin A, an inhibitor of the respiratory chain, slows down the growth of *S. pombe* cells in defined minimal medium, but much less so in rich yeast extract medium with the same glucose concentration (Malecki *et al*, 2016) (Fig EV1A). This finding suggests that the minimal medium lacks a supplement that becomes limiting when the respiratory chain is inhibited. Indeed, supplementation with soluble amino acids largely rescued this slow growth phenotype (Figs 1A and EV1A). This result indicates that respiratory inhibition leads to amino-acid auxotrophy that limits cell growth. To test whether any single amino acid is sufficient to rescue the slow growth during fermentation, we grew cells in minimal medium with antimycin A and supplemented with 18 individual amino acids; as a control, we used medium supplemented with the mix of all amino acids or without any supplementation. Notably, supplementation with arginine (R) alone could promote growth nearly as much as the mix of all amino acids (Fig 1A). This effect was dependent on the dose of arginine (Fig EV1B). Thus, arginine, but no other single amino acid, is sufficient to promote rapid cell growth when the respiratory chain is inhibited.

To test whether arginine is necessary for rapid cell growth with inhibited respiratory chain, we grew cells with all amino acids except arginine. While supplementation of the mix of all amino acids allowed rapid growth, omission of arginine from the mix led to slower growth similar to cells not supplemented with any amino acids (Fig 1B). We conclude that cell growth in minimal medium is reduced upon blocking the respiratory chain, and arginine supplementation is both sufficient and necessary to increase growth of respiring cells to near normal levels.

These results suggest that respiration is important for the synthesis of arginine. We therefore expected that respiration-defective mutants are auxotroph for arginine. To test this hypothesis, we screened for arginine-auxotroph mutants by growing the ∼ 3,500 strains of the haploid *S. pombe* gene deletion library in the presence or absence of arginine. This screen identified 31 mutants that grew substantially better when supplemented with arginine (Fig 1C; Dataset EV1). As expected, these arginine-auxotroph mutants included those with impaired arginine biosynthesis. Out of 11 genes associated with the Gene Ontology (GO) category "arginine biosynthesis" (GO:0006526), eight were identified in our screen (Fig 1C and D). Of the remaining three, *aca1* does not have a confirmed function in arginine metabolism, *arg7* is one of two argininosuccinate lyases (with deletion of the other one, *arg41,* showing arginine auxotrophy), and *arg3* is wrongly annotated in the deletion library, because an independently prepared deletion strain showed arginine auxotrophy (Fig EV4A). Notably, most steps of the arginine biosynthesis pathway occur in mitochondria (Fig 1D).

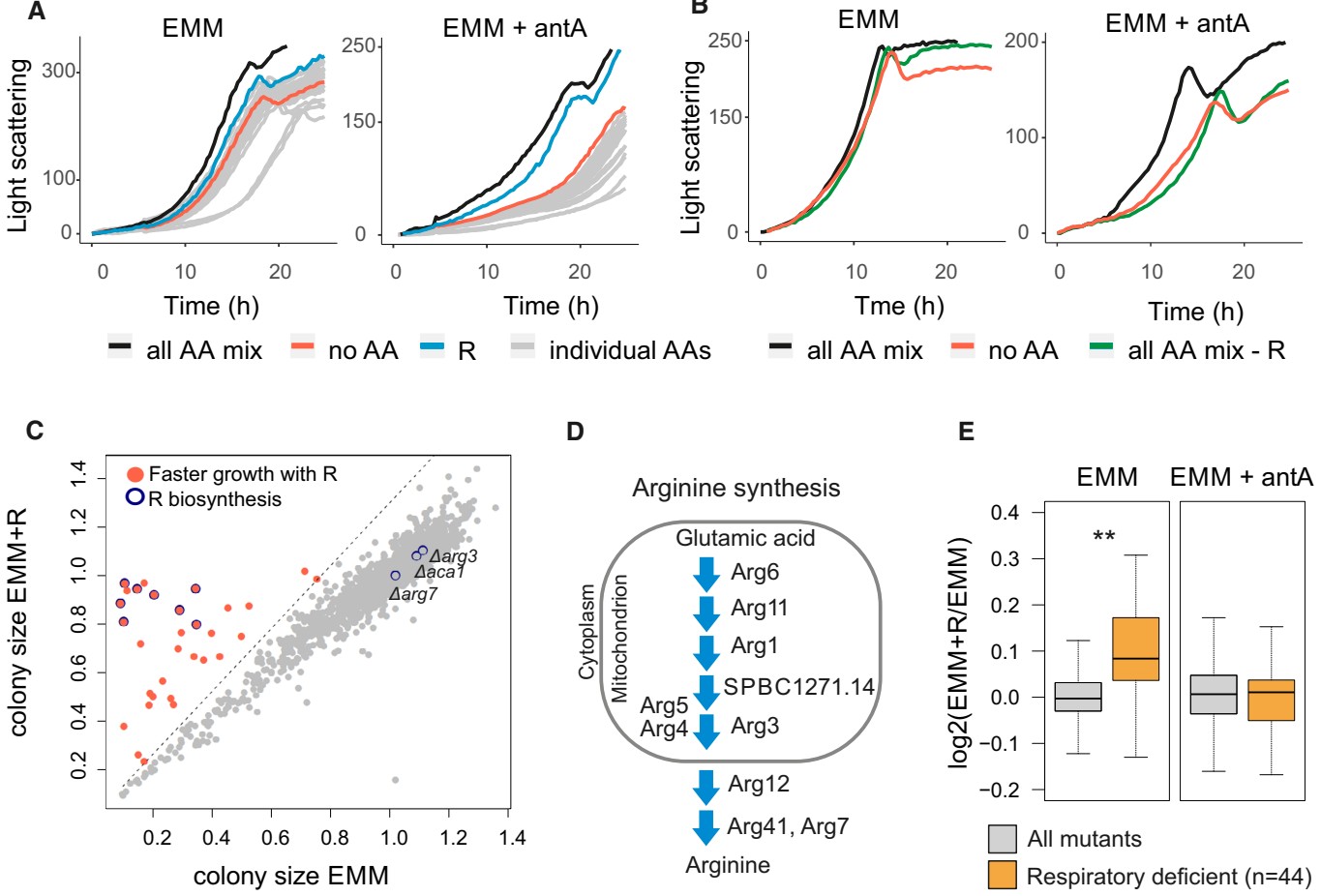

**Figure 1.  Arginine enhances growth of cells with inhibited respiratory chain.**

A   Growth curves of *Schizosaccharomyces pombe* cultures in minimal medium without (EMM) or with antimycin A (EMM + antA). Cultures were supplemented with either the complete amino-acid mix (all AA mix, black), without any amino acids (no AA, red), with arginine only (R, blue) or with individual amino acids other than arginine (individual AAs, grey).

B   Growth curves as in (A) for cultures supplemented with complete amino-acid mix (black), without any amino acids (red) or amino-acid mix without arginine (green).

C   Scatter plot of normalised colony sizes of cells grown on EMM against normalised colony sizes of cells grown on EMM supplemented with arginine (EMM + R; Dataset EV1). Mutants that passed our filtering (Materials and Methods) and whose colony size ratios were 30% bigger when supplemented with arginine are shown in red; mutants encoding known genes of the arginine biosynthesis pathway are shown with blue circles.

D   Scheme of arginine biosynthesis pathway, with protein localisation based on PomBase annotations (Lock *et al*, 2019).

E   Box plot of mean colony size ratios of mutants grown with *vs* without arginine for all mutants (grey) and respiratory deficient mutants (orange, based on previous screens (Malecki & Bähler, 2016)) (Welch's *t*-test **P < 0.002). The same analysis was performed for mutants grown on EMM (left; four biological repeats, Dataset EV1) or on EMM supplemented with antimycin A (right; two biological repeats, Dataset EV2). Boxplots were created using the R boxplot command with default settings. The central band indicates the median, the upper and lower limits of the box indicate first and third quartiles, and the whiskers show the most extreme data, limited to 1.5-time interquartile range.

The screen identified 23 additional genes not previously associated with arginine metabolism. Like the arginine biosynthesis genes, these 23 genes were enriched for those encoding mitochondrial proteins (13 of 23; *P* < 1E-5); some of these 13 genes encode mitochondrial metabolic enzymes (e.g. Lat1, Maa1, Idh1, Pos5, Atd1) whose deletion could impact arginine synthesis by affecting the Krebs cycle and glutamic acid production (Fig EV2A), while other genes encode respiratory functions like subunits of the mitochondrial ATPase or proteins involved in mitochondrial transcription (Ppr7) or translation (Mpa1). Our screen also identified a mutant of mitochondrial thioredoxin (Trx2) which is known to be rescued by arginine (Song *et al*, 2008). Threshold-free analysis of the screen

results further revealed significant enrichment of genes encoding mitochondrial proteins among the strains that grew better with arginine (Fig EV2B). Thus, arginine auxotrophy can be caused not only by mutants directly affecting arginine synthesis but also by mutants affecting different mitochondrial functions.

Given that arginine was not essential for cell growth after blocking the respiratory chain (Fig 1A), some respiratory mutants may show only subtle growth defects in the absence of arginine. Using data from our previous screen (Malecki & Bähler, 2016), we checked how a group of 44 respiratory deficient mutants grew in the screen reported here. Overall, the growth of respiratory mutants was significantly enhanced by arginine (Fig 1E). As a control, we screened all

mutants in the presence or absence of arginine but with antimycin A to block the respiratory chain in all strains. Under this condition, we did not detect any growth differences between respiratory and other mutants, indicating that growth inhibition without arginine was indeed caused by differences in respiration (Fig 1E; Dataset EV2). We also checked for effects of arginine on the growth of respiratory mutants in liquid medium: out of 44 respiratory mutants, 20 exhibited a growth phenotype in minimal medium (either slow growth or lower final biomass), while arginine supplementation attenuated the growth phenotype for 15 of these mutants (Fig EV3). We conclude that many respiratory mutants become dependent on arginine, indicating that mitochondrial respiration is important for arginine biosynthesis.

### Block of respiratory chain leads to changing cellular amino-acid levels

To further analyse the link between respiration and arginine metabolism, we measured the levels of free amino acids in cells treated with antimycin A. To this end, we used liquid chromatography selective reaction monitoring (Mülleder *et al*, 2017) to determine intracellular concentrations of 16 amino acids (Dataset EV3). Indeed, cells with blocked respiratory chain showed about ninefold reduced levels of arginine (Fig 2A). In addition, these cells showed reduced levels of ornithine (Orn), a metabolite of the arginine synthesis pathway, and of glutamic acid (E), a substrate for arginine biosynthesis (Ljungdahl & Daignan-Fornier, 2012) (Fig 2A). In contrast, levels of alanine (A) were elevated about 2.5-fold (Fig 2A). Interestingly, plants exposed to hypoxic conditions also accumulate alanine, which might serve as carbon/nitrogen reservoir (Diab & Limami, 2016).

We also measured the dynamic changes in free intracellular amino acids within 2 h after exposing cells to antimycin A. Consistent with cells grown in steady-state, inhibition of the respiratory chain led to increasing levels of alanine and decreasing levels of arginine (Fig 2B). Notably, several other amino acids decreased, mostly derivatives of Krebs cycle metabolites such as glutamic acid and glutamine. To confirm that the altered amino-acid levels are not an artefact of antimycin A treatment, we measured amino-acid concentrations in five respiratory mutants whose growth was improved by arginine (Fig EV3). In these mutants, except *mss1*, the patterns of amino-acid changes were similar to those of cells grown with antimycin A (Fig 2C). Most notably, the levels of arginine were lowered while the levels of alanine increased (Fig 2C). These results corroborate that the slow growth of respiratory mutants in minimal medium is caused at least in part by limiting intracellular levels of arginine. Together, these findings show that respiration is important for the synthesis of arginine and some other amino acids that are dependent on the Krebs cycle.

### Amino acids derived from arginine contribute to rapid cell growth with blocked respiratory chain

The changes in intracellular amino-acid concentrations pointed to additional effects from arginine supplementation: ornithine levels transiently increased, peaking 60 min after blocking the respiratory chain (Fig 2B), while they decreased after an extended respiratory block (Fig 2A). This result suggested that arginine synthesis may be blocked at the transition from ornithine to citrulline or that cellular

arginine is catabolised to ornithine. The first possibility is consistent with a report showing that absence of mitochondrial thioredoxin results in lowered Arg3 which catalyses the ornithine-to-citrulline transition (Song *et al*, 2008). In our hands, however, neither the localisation of Arg3 nor its protein or RNA levels did change after antimycin A treatment or in respiratory mutants (Fig EV4B–D). Arginine can be catabolised to other amino acids, a conversion that is highly efficient in *S. pombe* with arginine being rapidly catabolised to proline, glutamate, glutamic acid and lysine (Bicho *et al*, 2010). Accordingly, cells supplemented with arginine showed not only higher levels of arginine but also of amino acids produced by arginine catabolism: ornithine, proline and glutamic acid (Fig 3A). Thus, upon blocking of respiration and associated arginine synthesis, arginine catabolism may transiently increase the levels of ornithine (Fig 2B), which in turn is converted to other amino acids (Fig 3B). These findings raised the possibility that metabolites derived from arginine, besides the supplemented arginine itself, contribute to the rapid cell growth when the respiratory chain is inhibited (Fig 1).

To investigate the role of arginine catabolism after blocking the respiratory chain, we constructed three mutants deleted for the genes encoding the arginine catabolism enzymes Car1, Car2 and Car3. Like the wild-type cells, all three mutants showed reduced growth upon antimycin A treatment (Fig 3C). Arginine supplementation did enhance growth in wild-type and *car1* deletion cells, but not in *car2* and *car3* deletion cells (Fig 3C). This difference between the mutants may reflect that Car1 only plays a negligible role in arginine catabolism in standard conditions (Bicho *et al*, 2010). Our results show that arginine catabolism is required for rapid proliferation of arginine-supplemented cells with inhibited respiratory chain.

We checked what arginine-derived amino acids are required for rapid cell proliferation when blocking the respiratory chain. While no single amino acid could increase the growth of *car2* or *car3* mutants, a mix of all amino acids did increase the growth of both mutants (Fig 3D). To establish which amino-acid combinations are necessary to increase growth without respiration, we used the *car2* mutant which exhibited a stronger growth phenotype. Arginine combined with its catabolic products, glutamine, glutamic acid and lysine, led to full restoration of cell growth (Fig 3E). Of the three arginine-derived amino acids, glutamine and lysine led to stronger growth increases than glutamic acid. Taken together, these results show that respiratory chain inhibition does not result only in inhibition of arginine synthesis, but also in a shortage of glutamine, lysine and glutamic acid. All these amino acids are derivatives of alpha-ketoglutarate produced by the Krebs cycle, indicating that the reduced growth is caused by inhibition of the Krebs cycle. Thus, inhibition of the electron transport chain through antimycin A also triggers and inhibition of the Krebs cycle. This contrasts the situation in budding yeast where the Krebs cycle can be uncoupled from the activity of the electron transport chain (Kingsbury *et al*, 2015).

### Transcriptome regulation in response to respiratory chain block and arginine

Inhibition of the respiratory chain leads to a retrograde response, including the downregulation of genes for subunits of the electron transport chain (Epstein *et al*, 2001; Malecki *et al*, 2016). Given the impact of respiratory chain inhibition on amino-acid homeostasis (Fig 2), we used microarrays to investigate transcriptome changes

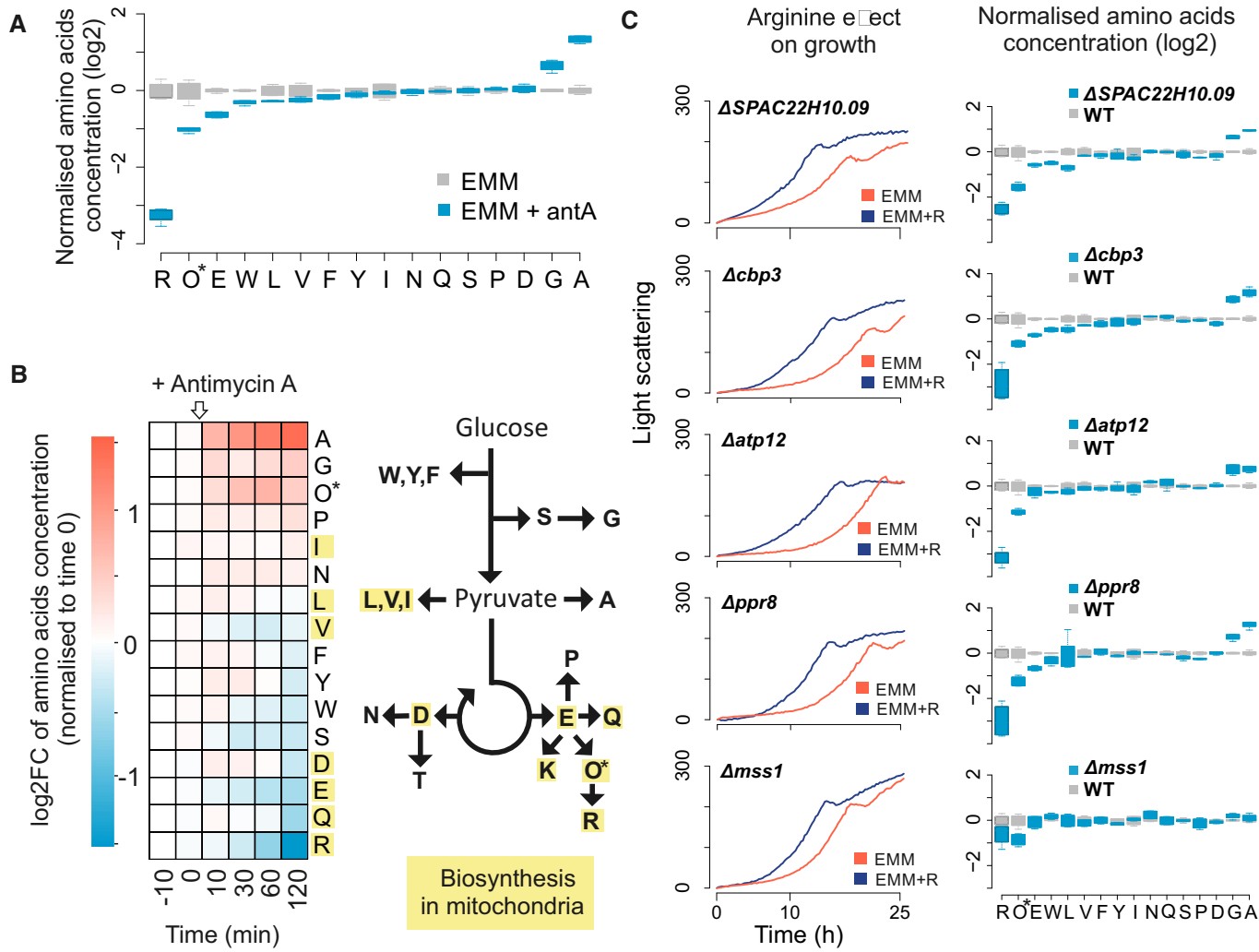

**Figure 2. Respiratory inhibition alters levels of cellular amino acids.**

A Normalised concentration values of 16 amino acids in cells grown in EMM without (grey) or with (blue) antimycin A. For each condition, four independent samples were analysed. Concentrations are presented relative to the mean value for the given amino acid in EMM medium. *Ornithine (Orn) is abbreviated as O. Boxplots were created using R boxplot command with default settings. The central band indicates the median, the upper and lower limits of the box indicate first and third quartiles, and the whiskers show the most extreme data, limited to 1.5-time interquartile range.

B Amino-acid concentrations before and after addition of antimycin A to cells exponentially growing in EMM. Left: heat map showing concentration changes (log$_2$ fold change) of each measured amino acid during the time course relative to time point 0 (before antimycin A addition). Right: simplified scheme of amino-acid biosynthetic pathways. Amino acids for which any biosynthetic enzyme has an annotated mitochondrial location are highlighted in yellow. *Ornithine (Orn) is abbreviated as O.

C Left panels: cell growth in EMM without (red) or with (blue) arginine supplementation for five selected respiratory mutants as indicated (graphs selected from Fig EV3). Right panels: normalised intracellular amino-acid concentrations as in (A) for the same five respiratory mutants (blue) and wild-type control cells (grey). For each mutant and the wild-type strain, four independent samples were analysed. *Ornithine (Orn) is abbreviated as O. Boxplot features as in (A).

within 2 h after antimycin A addition in the presence or absence of arginine. Addition of antimycin A inhibited the expression of genes functioning in the electron transport chain regardless of arginine supplementation (Fig EV5A). Moreover, antimycin A resulted in an immediate, strong core environmental stress response (CESR) (Chen *et al*, 2003), regardless of arginine supplementation (Fig 4A; Dataset EV4).

Interestingly, after addition of antimycin A, a group of 40 genes were less induced or even inhibited in the presence of arginine (Figs 4B and EV5B). Most of these genes are known to be induced after inhibition of TOR signalling (Figs 4C and EV5C) (Rodríguez-

López *et al*, 2019). Accordingly, this group was enriched for genes that are repressed or induced, respectively, upon genetic activation or inhibition of TOR kinase signalling (Fig 4D) (Van Slegtenhorst *et al*, 2004; Matsuo *et al*, 2007; Rallis *et al*, 2013). We therefore conclude that blocking the respiratory chain not only results in a stress response but also leads to a transient TOR inhibition and up-regulation of genes that are normally inhibited by TOR.

Arginine can activate TOR signalling (Bar-Peled & Sabatini, 2014; Davie *et al*, 2015); thus, the lower expression of the 40 genes in arginine-supplemented cells may reflect that arginine antagonized the TOR inhibition triggered by the respiratory chain block. This effect of

arginine could potentially explain its pro-growth effect when the respiratory chain is blocked. However, TOR-inhibited transcript genes were up-regulated within 30 min (Fig 4B), while arginine

concentration substantially decreased only 2 h after antimycin A treatment (Fig 2B). It is therefore more likely that arginine enhances growth of respiration-inhibited cells simply by replenishing the

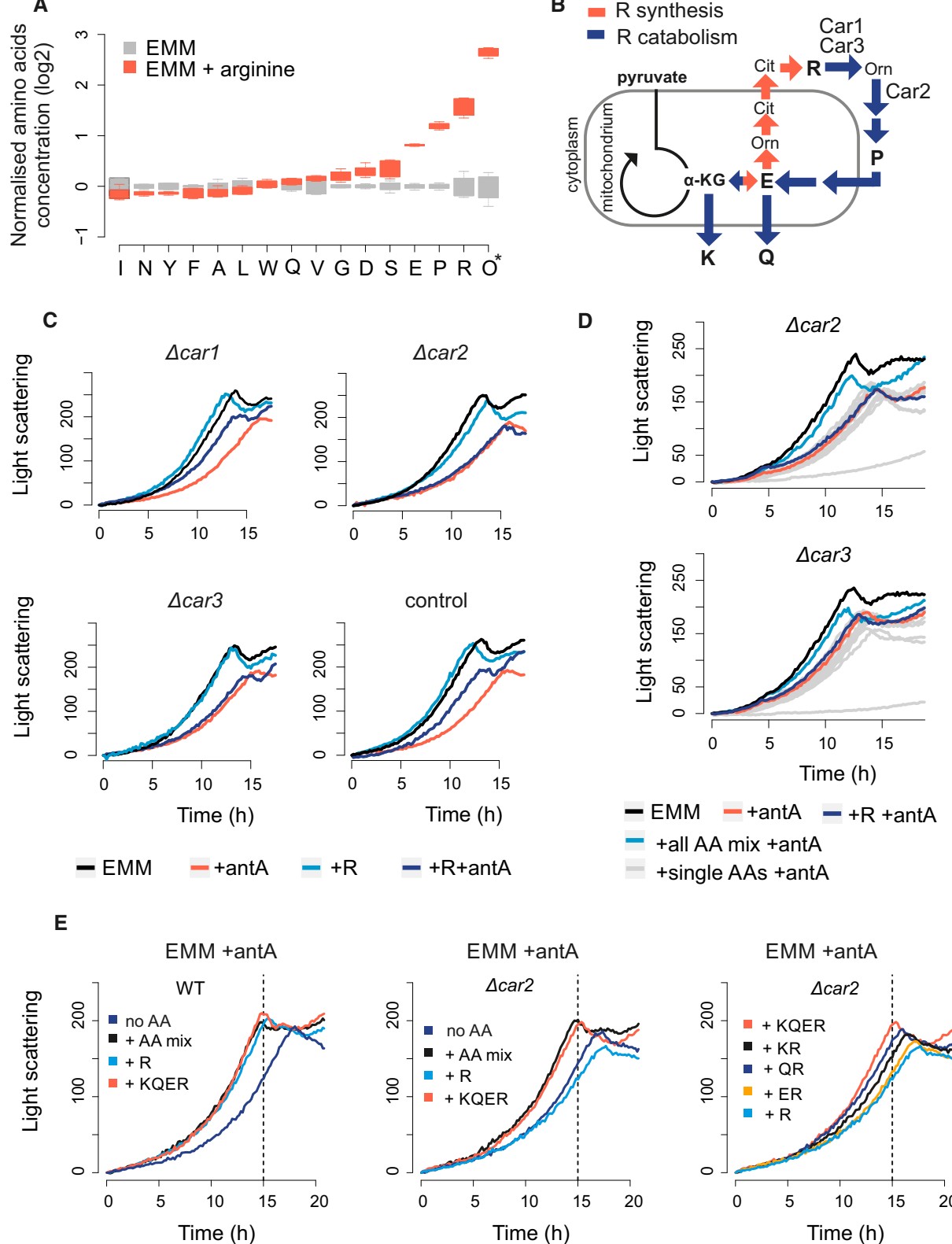

**Figure 3.**

**Figure 3.  Arginine catabolism contributes to growth of cells with blocked respiratory chain.**

A   Normalised amino-acid concentrations in cells grown in EMM without (grey) or with arginine (red). For each condition, four independent samples were analysed. Concentrations are presented relative to the mean value for the given amino acid in EMM medium. *Ornithine (Orn) is abbreviated as O. The central band indicates the median, the upper and lower limits of the box indicate first and third quartiles, and the whiskers show the most extreme data, limited to 1.5-time interquartile range.

B   Schematic relationship between alpha-ketoglutarate (α-KG) and arginine (R) synthesis (red arrows) and catabolism (blue arrows). Arginine is used to synthesise proline (P), glutamic acid (E), glutamine (Q) and lysine (K). The enzymatic roles of Car1, Car2 and Car3 are indicated.

C   Growth curves of strains deleted for *car1*, *car2* or *car3*, and a wild-type control. Strains were grown in EMM (black), EMM with antimycin A (red), EMM with arginine (light blue) or EMM with both arginine and antimycin A (dark blue) as indicated.

D   Growth curves of strains deleted for *car2* (top) or *car3* (bottom). Strains were grown in EMM (black), EMM with antimycin A (red), EMM with arginine and antimycin A (dark blue), EMM with mix of all amino acids and antimycin A (bright blue) and EMM with antimycin A and each individual amino acid from the mix (grey).

E   Growth curves of wild-type control (WT) and *car2* deletion strain (*Δcar2*). Strains were grown in EMM media with antimycin A and with either a mix of all amino acids (+ AA mix) or different combinations of lysine (K), glutamine (Q), glutamic acid (E) or arginine (R) as indicated. Dashed line: time point where wild-type strain saturates in EMM with amino-acid mix.

limiting amino acids. This conclusion is supported by our finding that supplementation with other amino acids that activate TOR, such as lysine or glutamine, did not promote cell proliferation (Fig 1A).

Given that the rapid, transient TOR inhibition is unlikely to result from the lower arginine levels in response to a respiratory inhibition, it may be triggered by a different signal. We hypothesised that the AMP-activated protein kinase (AMPK) could mediate such an earlier signal. Consistent with this idea, antimycin A-treated cells without the AMPK-activating kinase Ssp1 did not up-regulate most of the 40 genes that were repressed by arginine (Fig 4C). This result suggests that TOR is indeed inhibited by AMPK kinase upon blocking respiration. Consistently, regulation of TOR via AMPK in response to environmental perturbations has been reported in fission yeast (Davie *et al*, 2015; Forte *et al*, 2019), and lowering the Krebs cycle flux leads to AMPK-mediated TOR inhibition in budding yeast (Kingsbury *et al*, 2015). After blocking respiration, the cells may bounce back from the transient TOR inhibition, through a retrograde response and redirection of energy metabolism to fermentation, until the levels of arginine and derived amino acids become limiting for rapid proliferation.

The protein products of the 40 TOR-regulated genes that are repressed by arginine supplementation (and induced by antimycin A) featured lower than average contents of nitrogen, arginine, lysine, glutamine and glutamic acid (Fig 4E). Strikingly, these amino acids are all derived from alpha-ketoglutarate which becomes scarce without sufficient flux in the Krebs cycle (Fig 2B). This finding illustrates the relationship between available resources and cell regulation. So a block in the respiratory chain results in shortage of amino acids derived from alpha-ketoglutarate, and proteins that are induced to deal with this shortage, including amino-acid transporters; (Fig EV5C), feature low levels of these limiting amino acids. On the other hand, these induced proteins featured higher than average contents of alanine, isoleucine, valine, glycine and threonine, consistent with the intracellular increase in alanine and glycine in the absence of respiration (Fig 2B). These findings indicate that the gene expression response to altered levels of arginine or respiration involves proteins whose content is adapted to intracellular amino-acid availability in these conditions.

## Conclusions

We demonstrate here that inhibiting the respiratory chain generates a fitness cost for *S. pombe* cells grown in fermentative conditions.

Respiration contributes to rapid proliferation during aerobic fermentation by supplying amino acids derived from alpha-ketoglutarate, a Krebs cycle metabolite. Respiratory inhibition therefore leads to a shortage of arginine, glutamic acid, glutamine and lysine. This shortage can be largely alleviated by supplementation with arginine which is efficiently converted to other alpha-ketoglutarate derivatives. Our transcriptome analysis shows not only a retrograde response but also a transient, rapid response to stress and TOR inhibition after blocking the respiratory chain. An intact AMPK pathway is required for this TOR inhibition. Budding yeast features a similar response to a Krebs cycle block, but not to a respiration block (Kingsbury *et al*, 2015). Our screen for mutants whose growth is improved by arginine revealed several strains that cannot grow on non-fermentable carbon sources, including mutants of mitochondrial ATP synthase. The finding that not all respiratory deficient strains exhibit detectable arginine auxotrophy suggests that arginine levels depend on secondary effects of the respiration block such as disruption of the mitochondrial membrane potential.

Some fission yeast isolates, including the standard laboratory strain, possess a mutation in pyruvate kinase, a key glycolytic enzyme (Kamrad *et al*, 2020). This feature limits glycolytic flux and causes increased respiration during fermentative growth (Kamrad *et al*, 2020). It is possible that the shortage of amino acids is exacerbated by this mutation. *Schizosaccharomyces pombe* thus provides a unique model to study how increased respiration compensates for lower glycolytic flux. Rapid proliferation is usually supported by fermentative metabolism even in the presence of oxygen. However, aerobic glycolysis does not replace respiration, and it becomes clear that both processes occur in parallel to support the metabolic needs of cells. We show here that respiration in *S. pombe* is required to supply amino-acid derivatives of the Krebs cycle. Arginine is necessary and sufficient to suppress the growth inhibition triggered by blocking the respiratory chain.

While budding yeast exhibits several adaptations to growth without respiration, petite strains without mitochondrial DNA still show a slow growth phenotype (Ephrussi *et al*, 1949). Budding yeast cells with compromised respiration show altered amino-acid profiles (Mülleder *et al*, 2016; Druseikis *et al*, 2019) but no obvious auxotrophies have been reported. In human cells, respiration limits proliferation due to its contribution to aspartate synthesis (Birsoy *et al*, 2015; Sullivan *et al*, 2015). Arginine is semi-essential in humans and is an essential supplement for mammalian cell cultures, rendering it difficult to study the impact of respiration on arginine levels in this system. Intriguingly, several reports indicate positive effects of

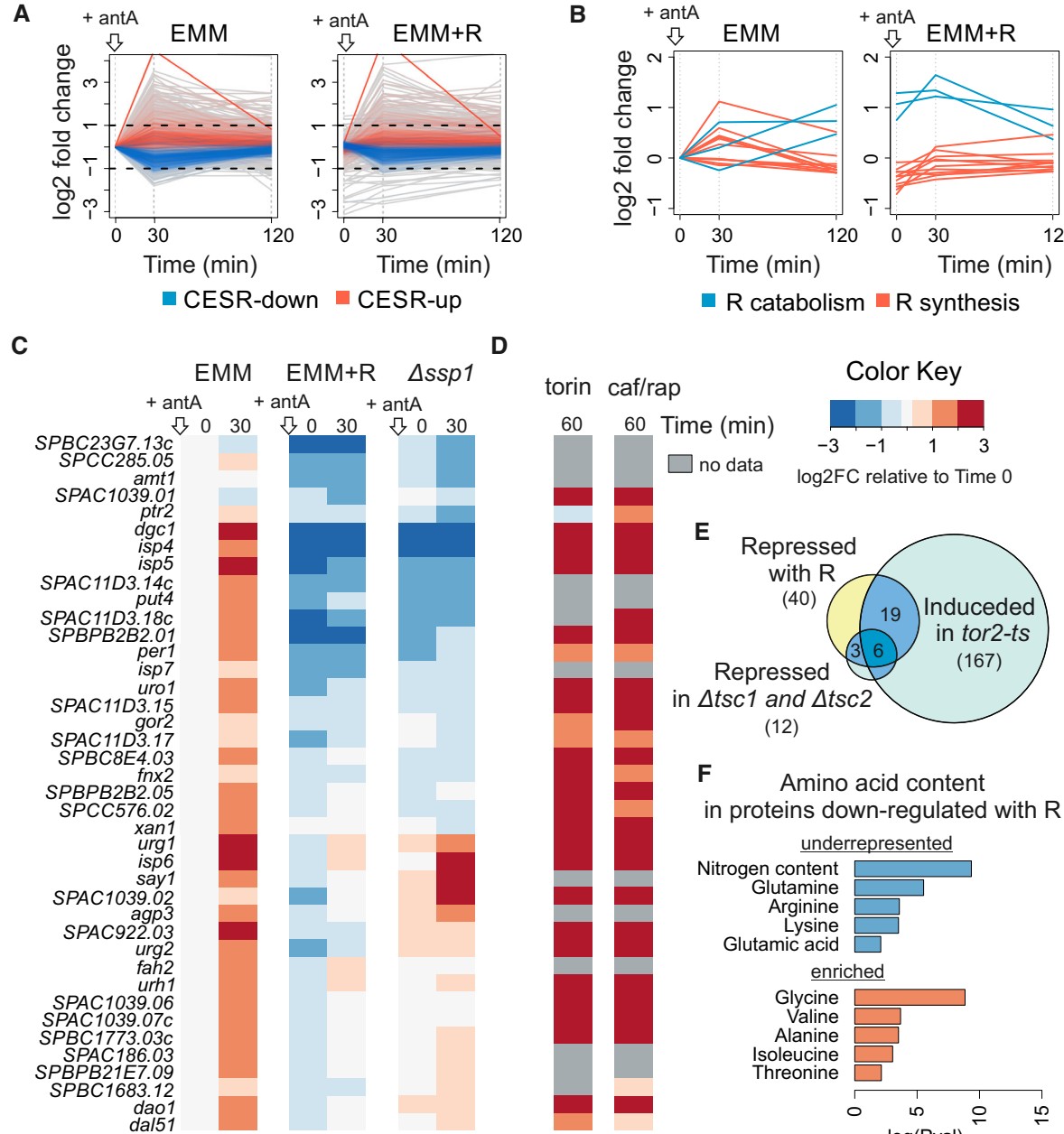

**Figure 4. Transcriptome regulation in response to respiratory inhibition and arginine supplementation.**

A Changes in transcript abundance 30 and 120 min after addition of antimycin A to exponentially growing cells without (EMM) or with arginine (EMM + R). Transcripts annotated as up- or down-regulated core environmental stress response (CESR) are indicated in red and blue, respectively. All data are normalised to EMM time point 0.

B Graphs as in (A) showing expression profiles of transcript encoding enzymes for arginine catabolism (blue) or arginine biosynthesis (red).

C Heat map of expression changes for transcripts repressed over 2-fold at 30 min in cells supplemented with arginine (EMM + R) relative to cells without supplementation (EMM). Values for timepoints 0 and 30 min after antimycin A addition are shown for cells grown in EMM, EMM with arginine and *ssp1* deletion mutant grown in EMM. The data are normalised to EMM timepoint 0.

D Heat map of expression changes of the same genes as in (B) after 1 h of inhibiting TOR kinase with torin1 or caffeine/rapamycin treatment relative to cell before treatment; cells were grown in rich media—YES (data from (Rodríguez-López *et al*, 2019)).

E Overlaps between genes repressed with arginine supplementation and genes induced in *tor2* thermo-sensitive mutant (*P* < E-27) (Matsuo *et al*, 2007), or repressed in strains deleted for the TOR inhibitors *tsc1* and *tsc2* (*P* < E-16) (Van Slegtenhorst *et al*, 2004).

F Enriched and underrepresented amino acids in proteins encoded by 40 genes that are down-regulated in media with arginine. Analysis performed using online tool AnGeLi (Bitton *et al*, 2015).

arginine supplementation on patients with mitochondrial dysfunction (Ganetzky & Falk, 2018).

## Materials and Methods

### Yeast strains and growth media

Strains used in this study are listed in Table EV2. For colony size measurements, the auxotroph Bioneer library v5.0 was used (Kim et al, 2010). Respiratory mutants were isolated from prototroph derivative of auxotroph collection (Malecki & Bähler, 2016). Cells were grown in rich Yeast Extract with Supplements (YES) medium with 3% glucose or defined Edinburgh Minimal Medium (EMM) with 2% glucose (Formedium PMD12CFG). Antimycin A was used where indicated at 0.15 µg/ml concentration. Amino acids were used at 0.2 g/l concentration each unless indicated otherwise.

### Growth recording and analysis

Growth curves of cells growing in liquid media were recorded using BioLector (m2P-laboratories) microfermentor. Data were normalised to time zero and visualised using custom R scripts.

For colony size measurement, deletion library was arrayed on solid agar EMM media plates in 384 colony/plate format using a RoToR HDA robot (Singer Instruments). After 2 days of growth, plates were scanned and raw colony sizes extracted using gitter R package (Wagih & Parts, 2014). Colony sizes were normalised using an in-house R package (HDAr package—unpublished). Plates were supplemented with arginine or antimycin A. Additionally, plates were supplemented with adenine, uracil and leucine. Experiment with arginine supplementation was performed in four repeats (two parallel repeats in two different batches). Experiment with arginine supplementation in presence of antimycin A was performed in two parallel repeats. All normalised colony sizes from aforementioned experiments are provided in Dataset EV1 and Dataset EV2. To identify strains growing better on arginine, we extracted colony size information from all EMM and EMM + R plates and removed data for most variable colonies for each condition (coefficient of variation > 0.25).

### Amino acids measurement

Intracellular amino acids were quantified as previously described (Jeffares et al, 2015; Mülleder et al, 2016, 2017). In brief, samples were prepared by hot ethanol extraction of mid-log phase cultures. Then, 1.5 ml l of culture was spun down, washed with deionised water (3,000 g, 3 min), and 200 µl of ethanol at 80°C was added. The pellet was resuspended by vortexing and incubated for 2 min at 80°C, followed by vortexing and another 2-min incubation step. The extract was cleared by centrifugation at 3,000 g for 5 min, and the supernatant was used for LC-MS analysis without further conditioning. Chromatographic separation was performed on 1 µl of sample using a Waters ACQUITY UPLC BEH amide 1.7 mm 2.1 × 100 mm column at 25°C. Flow was maintained at 0.9 ml/min, starting with 15% Buffer A (50:50 acetonitrile/water, 10 mM ammonium formate, 0.176% formic acid) and 85% Buffer B (95:5:5 acetonitrile/methanol/water, 10 mM ammonium formate and 0.176% formic acid) for 0.7 min, followed by a 1.85-min ramping to 5% B, and

keeping constant for 0.05 min before returning to the equilibrating conditions of 85% B (total runtime 3.25 min). Quantification was carried out by multiple reaction monitoring (MRM) on a triple quadrupole mass spectrometer (Agilent 6460). Peak areas were converted to concentrations by external calibration with amino acid standard mixtures. Cysteine, methionine, lysine and histidine could not be resolved or quantified reliably and were excluded from further analysis. Data were normalised by median quotient normalisation as described (Dieterle et al, 2006).

### Microarray experiments

For the time course analyses, cells were grown to early exponential phase (OD 0.5) in EMM medium with or without 0.2 g/l arginine. Transcriptomes were analysed before (time point 0) and at two time points after adding antimycin A to the culture (30 and 120 min). Cells were collected using centrifugation, washed once with ice-cold water and snap-frozen. RNA from cell pellets was isolated using hot phenol extraction, followed by labelling of the single samples and a pool of all the samples which served as reference (Lyne et al, 2003). Agilent 4 × 44K custom-made S. pombe expression microarrays were used, and hybridizations and subsequent washes were performed according to the manufacturer's protocols. Microarrays were scanned using a GenePix 4000 B laser scanner, and fluorescence signals were analysed using GenePix Pro software (Axon Instruments). The resulting data were processed using customised R scripts for quality control and normalisation and analysed using GeneSpring software. Normalised data from GeneSpring are included in Dataset EV4. All the visualisations were performed based on this data set with use of custom R scripts.

## Data availability

Raw microarray data are accessible via ArrayExpress, accession number: E-MTAB-8805 (http://www.ebi.ac.uk/arrayexpress/experiments/E-MTAB-8805).

Expanded View for this article is available online.

### Acknowledgements

We thank Maria Rodríguez-López for help with microarray analysis, Cristina Cotobal for sharing unpublished results, StJohn Townsend for help with colony size screen analysis, Mimoza Hoti for technical support and Charalampos Rallis for advice and fruitful discussions. We thank Shajahan Anver, Snezhka Oliferenko, Charalampos Rallis and Maria Rodríguez-López for comments on the manuscript. This work was supported by a Wellcome Trust Senior Investigator Award to J.B. (grant no. 095598/Z/11/Z), by the Foundation for Polish Science FIRST TEAM grant awarded to M.M (POIR.04.04.00-00-4316/17), and by the Francis Crick Institute which receives its core funding from Cancer Research UK (FC001134), the UK Medical Research Council (FC001134) and the Wellcome Trust (FC001134).

### Author contributions

MM and JB conceived this work. MM designed and executed most of the experiments. SK performed the mass spectrometry analyses. SK and MR provided valuable suggestions. MR and JB secured the research funding. MM drafted the manuscript. MM and JB, with contributions from all authors, prepared the final version of the manuscript.

## Conflict of interest

The authors declare that they have no conflict of interest.

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
