## [Review Process File · EMBO Reports]

Mitochondrial respiration is required to provide amino acids during fermentative proliferation of fission yeast

Michal Malecki, Stephan Kamrad, Markus Ralser, and Jürg Bähler

DOI: doi.org/10.15252/embr.202050845

Corresponding authors: Jürg Bähler (j.bahler@ucl.ac.uk), Michal Malecki (m.malecki@ucl.ac.uk)

Review Timeline:	Transfer from Review Commons:	8th May 20
	Editorial Decision:	12th Jun 20
	Revision Received:	7th Jul 20
	Accepted:	10th Aug 20

Editor: Deniz Senyilmaz Tiebe

Transaction Report: This manuscript was transferred to EMBO reports following peer review at Review Commons

Review #1

1. How much time do you estimate the authors will need to complete the suggested revisions:

Estimated time to Complete Revisions (Required)

(Decision Recommendation)

Between 1 and 3 months

2. Evidence, reproducibility and clarity:

Evidence, reproducibility and clarity (Required)

In most animals, arginine is an essential amino acid. In contrast, most fungi are arginine prototrophs. In the present study, the authors report about the relevance of respiration for arginine synthesis in *Schizosaccharomyces pombe*. Moreover, they observed that arginine supplementation can reduce the growth defects of some respiratory mutants. In a genetic screen, they identify genes which are critical for arginine prototrophy. These genes contain the enzymes of the arginine synthesis pathway as well as some components of the TCA cycle and NAD(P) oxidoreductases. Finally, they report about a link between arginine availability, respiration and TOR signaling. The study is clear and good to read, the experiments are of high quality and the necessary controls are shown. However, the relevance of respiration for arginine biosynthesis is not novel. A detailed study on this connection was published some time ago in *Current Biology* (Druseikis M, Ben-Ari J, Covo S. The Goldilocks effect of respiration on canavanine tolerance in *Saccharomyces cerevisiae*. *Curr Genet*. 2019; 65: 1199-1215) in a study that is not even mentioned in this manuscript. This certainly limits the relevance of this study, despite the high technical quality. **Major points**

1. The results need to be discussed and described in comparison to the Druseikis study.
2. In their arginine screen, the authors identified several components that are critical for the redox homeostasis in the mitochondrial matrix, such as Pos5 or Trx2. This points at a relevance of arginine synthesis on the redox homeostasis in the mitochondrial matrix. The study would be considerably stronger if the authors would have employed redox probes, such as those developed by Tobias Dick, to elucidate the relevance of arginine levels on the glutathione redox potential or the levels of hydrogen peroxide in the mitochondrial matrix.
3. **Minor points**
3. The nomenclature of the enzymes for arginine synthesis in *S. pombe* obviously differs from that in *S. cerevisiae*. The authors should add a table to Fig. S2 that allows to compare the names of genes identified here to gene names of *S. cerevisiae*.
4. In the screen for arginine auxotrophic mutants, the authors identified proteins essential for respiration such as Atp1. How can these mutants survive in a petite-negative organism such as *S. pombe*? Why were other ATPase mutants not found?
5. The tables shown in the supplemental excel files do only list systematic gene names. They are basically unreadable for most readers unless they invest some work. The authors should add protein names to these tables.

3. Significance:

Significance (Required)

The data are of very high quality. However, most of this study is confirmatory as a similar study was published recently. The screen for arginine auxotrophy is of interest for a small group of people studying amino acid biosynthesis in *S. pombe*.

Review #2

1. How much time do you estimate the authors will need to complete the suggested revisions:

Estimated time to Complete Revisions (Required)

(Decision Recommendation)

Less than 1 month

2. Evidence, reproducibility and clarity:

Evidence, reproducibility and clarity (Required)

The studies in this manuscript address the question as to why fission yeast partially rely on respiration for rapid proliferation even in the presence of a fermentable carbon source. The study addresses this question with a series of experiments which meticulously delineates the factors. In the first study, it is shown that supplementation of the growth medium with arginine alone is sufficient to relieve the need for respiration for rapid proliferation. They then determined, using the collection of yeast null deletion mutants, that loss of function of a number of genes mimics effect by the loss of respiration and the effect of arginine. Many of these genes were in the arginine pathway, as expected, and many were in the respiratory chain or in those related to the ATP synthase. In the last study, the authors investigate gene expression profiles in response to a block in respiration in the presence and absence of arginine. These data indicate that there is also an effect on the genes involved in regulation of TOR. The principal conclusion of these studies is that in fission yeast, the respiratory chain is needed for rapid cell growth due to the role of the TCA in amino acid synthesis - specifically, that of Arg and its metabolic products. Overall, this is a well controlled study that makes sound conclusions. The main issues I have are related to Fig. S3 where genes are identified that when deleted, make the cells respond to Arg supplementation. In some cases, the discrimination factor is small where just a small effect is seen - as with *shy1* and *rrf1*. In other cases, the effects are not equivalent despite the fact that they are related to the same pathway or enzyme. For instance, the effect by loss of *atp1* is much smaller than that of *atp12* or *atp7* and *atp2* is not even listed. While this is not significant enough to prevent from publication, it does suggest that something more is

happening. Also, why are mutation in genes encoding the ATP synthase related to the electron transport chain? A minor correction - I think - is on p.7, last para, "... although arginine is essential in humans..." should be "... although arginine is not essential in humans..."

3. Significance:

Significance (Required)

The findings in this study are significant if only related to yeast metabolism, but highly significant if it can be related to mammalian metabolism. There are a number of human conditions where mitochondrial function is important and the loss there-of is detrimental. On the surface, this seems to be an obvious statement, but it is much less clear. For instance, many cancer cells grow without usage of the respiration - referred to as the Warburg effect. Thus, a priori, cells do not need the respiratory chain for ATP synthesis. There is loss of mitochondrial function with aging and in many neurodegenerative diseases. The studies here provide a possible link for the need of respiratory chain with cell growth; it is unclear how this will be related to human cells. Outside the references listed in the discussion of this manuscript, there is some relationship of this work with the studies on the genetics and biochemistry of "petite negative" yeasts. These studies looked to address why some yeast cannot lose their mitochondrial DNA while "petite positive" yeast can lose their mitochondrial DNA. The study here is the first study to provide a possible link to understanding this dichotomy. The audience that would be interested in this work are likely to be those that study yeast mitochondria metabolism and genetics. However, if this study furthered and related to human cells or diseases, then this can be a high impact paper. My expertise is more in yeast genetics, mitochondria, ATP synthesis, and protein structure. I can't say that I am expert on any specific aspect in this report, but I am confident that I understand the rationale, methods, and results.

Review #3

1. How much time do you estimate the authors will need to complete the suggested revisions:

Estimated time to Complete Revisions (Required)

(Decision Recommendation)

Between 1 and 3 months

2. Evidence, reproducibility and clarity:

Evidence, reproducibility and clarity (Required)

This is an interesting study that describes the metabolic contribution of the Krebs cycle to fission yeast growth under fermentative conditions (i.e. with glucose as carbon source). The authors build on their observation that chemical interference with the electron transport chain with antimycin has strong effects on cell growth in glucose minimal medium but much less effect in rich medium with glucose as carbon source. The authors then convincingly show that this effect is a specific result of an arginine deficit in the challenged cells, and a genome-wide screen reinforces the conclusion by showing a strong correlation between arginine 'auxotrophy' and defects in mitochondrial functions. The primary conclusion of the work is that fission yeast respiration compensates for arginine deficits resulting from fermentative growth, that this compensation translates to alleviation of deficits in arginine-derived amino acids as well (primarily glutamine, lysine and glutamate), and that these deficits are correlated with but independent of TOR regulation. In sum, this is a nicely comprehensive and systematically constructed body of work, and it is logically consistent across the multiple independent lines of investigation that were followed. The unexpected but attractive finding that further elevates the impact of this MS is that the proteins induced by arginine deficit (as compensatory mechanism) are underrepresented in arginine, lysine, glutamine and glutamate. Overall, a well-done analysis that addresses an important, but under-investigated, area of cellular physiology.

3. Significance:

Significance (Required)

The major significance of this work is that it addresses an important, but under-investigated, area of cellular physiology. That is, the crosstalk between glycolysis and the respiratory chain under conditions of aerobic glycolysis. The demonstration that this crosstalk focuses down on the specific target of arginine availability, and consequently to the availability of amino acids derived from arginine (lysine, glutamine, glutamate) is interesting. Of particular interest is the fact that, while these amino acids activate TOR, and the TOR pathway is inhibited by the respiration block in aerobically fermenting cells, that relationship is correlative yet independent. That the gene products induced to compensate for these amino acid deficits are themselves underrepresented This general question is of significant contemporary interest and it will be of interest to a broad audience that will include researchers in the biochemistry, cell cycle and cancer cell biology fields. The manuscript is clearly written and the conclusions well-supported. I pose three issues for consideration by the authors: (i) The manuscript would be strengthened by a flux experiment where mass-tagged arginine is fed to antimycin-treated cells and the rate of its conversion to lysine, glutamine and glutamic acid is followed in a time course. Those data could then be correlated to the existing transcriptomic data. This experiment would provide a direct measurement of the kinetics of conversion which is otherwise inferred from the existing data. Since the readout used is an LC-MS measurement, this should be a straightforward experiment. (ii) What happens to the arginine auxotrophy in antimycin-treated cells if the natural pyruvate oxidase mutation in fission yeast is corrected? Presumably, this correction would rescue the arginine auxotrophy. (iii) Why did the *mss1* mutant fail to recapitulate the amino acid deficiencies seen in antimycin-treated cells and in the four other respiratory chain mutants analyzed?

Dear Editor,

We thank the three reviewers for the constructive feedback. We have addressed the issues raised by text changes which have helped to improve the manuscript. Below we address the specific concerns with detailed responses.

Reviewer #1 (Evidence, reproducibility and clarity (Required)):

*In most animals, arginine is an essential amino acid. In contrast, most fungi are arginine prototrophs. In the present study, the authors report about the relevance of respiration for arginine synthesis in *Schizosaccharomyces pombe*. Moreover, they observed that arginine supplementation can reduce the growth defects of some respiratory mutants. In a genetic screen, they identify genes which are critical for arginine prototrophy. These genes contain the enzymes of the arginine synthesis pathway as well as some components of the TCA cycle and NAD(P) oxidoreductases. Finally, they report about a link between arginine availability, respiration and TOR signaling. The study is clear and good to read, the experiments are of high quality and the necessary controls are shown.*

Reply: We thank the Reviewer for this positive assessment.

*However, the relevance of respiration for arginine biosynthesis is not novel. A detailed study on this connection was published some time ago in *Current Biology* (Druseikis M, Ben-Ari J, Covo S. The Goldilocks effect of respiration on canavanine tolerance in *Saccharomyces cerevisiae*. *Curr Genet.* 2019; 65: 1199-1215) in a study that is not even mentioned in this manuscript. This certainly limits the relevance of this study, despite the high technical quality.*

Major points

1. The results need to be discussed and described in comparison to the Druseikis study.

Reply: We apologize that this recent paper by the Covo lab has not been cited. This paper, published in *Current Genetics*, focuses on canavanine resistance rather than amino acid metabolism. It makes a good addition to our discussion, as it establishes a link of canavanine toxicity and respiratory deficiency [1]. We have now cited and discussed this paper.

We strongly disagree, however, that the paper by the Covo lab compromises the impact or novelty of our study. **First**, their study is conducted in budding yeast, which does not require respiration during exponential growth on glucose, the very phenomenon we analyze in our paper. The paper by Druseikis et al. makes not even a reference to this phenomenon. **Second**, the main point of our study is not the relevance of respiration for arginine synthesis (as suggested by the Reviewer), but that respiration is required for rapid proliferation in fermenting cells by boosting the supply of Krebs-cycle derived amino acids. It has been proposed by Liu & Butow in 1999 [2] that petites feature compromised production of amino-acid derivatives of 2-ketoglutarate, and a high-throughput study by Mülleder et al. [3] provides information about amino-acid levels in all non-essential budding yeast deletion mutants, including respiratory mutants. Thus, we are well-aware that in budding yeasts amino acids, including arginine, change with compromised respiration, and we have indeed cited this previous evidence in our manuscript. However, unlike in fission yeast, arginine is not among the most affected amino acids in budding yeast (see **Figure 1** below). **Third**, Druseikis et al. provide no metabolic-mechanistic explanation for the connection of arginine metabolism and respiration. Druseikis et al. focus on the impact of canavanine on cells with different metabolic regimes, on protein localisation, and on membrane potential and the retrograde response. This paper makes hence no contribution to answering the problem we answer in our manuscript: why fission yeast – in contrast to budding yeast – needs respiration during exponential fermentative proliferation on glucose. Notably, most other organisms seem to behave like fission yeast in this respect.

Budding yeast seems to have evolved to prevent extensive amino-acid shortages in the absence of respiration (budding yeast are facultative anaerobes and petite positive cells unlike the evolutionary distant fission yeast). Petite cells, although compromised for amino-acid production, do not exhibit obvious auxotrophies (which is also noticed by Druseikis et al.). In fission yeast, on the other hand, a respiration block leads to obvious growth inhibition due to amino-acid shortage. We therefore provide solid evidence that amino-acid synthesis is an additional constraint for cellular energy metabolism.

We propose that also in multicellular organisms this possibility should be considered, which is relevant for models describing constraints that impact energy production and help us to understand the metabolic needs of mammalian cells using fermentative metabolism, like cancer cells.

Figure 1. Respiratory deficiency in budding results in altered amino-acid profiles, with arginine being less affected than in fission yeast.

(A) Distribution of normalised amino-acid concentrations in all budding yeast deletion mutants (orange; n = 4678) vs mutants from GO category “respiratory growth absent” (blue; n = 570). Amino acids are sorted based on the difference of median concentration between the two tested lists.

(B) GSEA enrichment scores for GO category: “mitochondrial translation” calculated for *S. cerevisiae* mutants ranked based on amino-acid concentrations (original data from ref. [3]).

2. In their arginine screen, the authors identified several components that are critical for the redox homeostasis in the mitochondrial matrix, such as *Pos5* or *Trx2*. This points at a relevance of arginine synthesis on the redox homeostasis in the mitochondrial matrix. The study would be considerably stronger if the authors would have employed redox probes, such as those developed by Tobias Dick, to elucidate the relevance of arginine levels on the glutathione redox potential or the levels of hydrogen peroxide in the mitochondrial matrix.

Reply: We agree that this could be a worthwhile direction of future research, although it is somewhat tangential to the focus of the current manuscript. At this time, we are not in a position to perform any additional experiments due to restrictions caused by the pandemic. More importantly, we believe that our current manuscript is well-focussed and supports coherent conclusions. Large-scale data are typically best interpreted through functional enrichments of gene groups rather than focussing on cherry-picked single genes. Notably, redox homeostasis did not show up as a functional category that was significantly enriched among genes that grow better with arginine, which makes the proposed hypothesis less attractive to pursue.

****Minor points****

3. The nomenclature of the enzymes for arginine synthesis in *S. pombe* obviously differs from that in *S. cerevisiae*. The authors should add a table to Fig. S2 that allows to compare the names of genes identified here to gene names of *S. cerevisiae*.

Reply: We have added an additional table to Figure S2 with a short description of the known functions of genes shown in the figure as well as the names of their budding yeast and human orthologues.

4. In the screen for arginine auxotrophic mutants, the authors identified proteins essential for respiration such as *Atp1*. How can these mutants survive in a petite-negative organism such as *S. pombe*? Why were other ATPase mutants not found?

Reply: Fission yeast is a petite-negative organism, meaning that it does not tolerate loss of mitochondrial DNA. But many respiratory mutants are viable including the *atp1* deletion, according to our and other data: <https://www.pombase.org/gene/SPAC14C4.14>. It is known that fission yeast

can survive without respiration. It has been suggested that loss of mitochondrial DNA is lethal because it compromises the functionality of both the electron transport chain and mitochondrial ATPase, and in these conditions fission yeast cannot restore mitochondrial membrane potential [4][5]. Mutations in ATP1 and ATP3 genes were found to suppress the petite-negative phenotype and allow the petite-negative yeast *K. lactis* to survive without mtDNA [5]. Suppressive mutations allowing for mtDNA loss were also described in fission yeast [5], but their nature is unknown as so far.

5. The tables shown in the supplemental excel files do only list systematic gene names. They are basically unreadable for most readers unless they invest some work. The authors should add protein names to these tables.

Reply: Common gene names (where available) as well as short descriptions of gene functions and budding yeast and human orthologues have been added to Tables S1 and S4.

Reviewer #1 (Significance (Required)):

The data are of very high quality. However, most of this study is confirmatory as a similar study was published recently.

Reply: As discussed in detail above, the study from the Covo lab does not address the problem of why *S. pombe* relies on respiration during exponential growth. It studies canavanine toxicity in *S. cerevisiae*. The two studies superficially relate in the sense that both depend on a link between arginine and mitochondrial function, but we do not see that why two studies about two different problems in two different organisms would be redundant. Our study is not at all similar to the study by Druseikis et al.

The screen for arginine auxotrophy is of interest for a small group of people studying amino acid biosynthesis in S. pombe.

Reply. We respectfully disagree that studying the respiration-fermentation balance is of interest for only a small group of people. The biotechnological and agricultural industry involving yeast fermentation is a multi-billion dollar business, involving >1,000,000 employees worldwide. Each process, from making bread, alcoholic beverages to producing anti-malaria drugs, depends on the tuning of fermentation processes, and the efficiency of the process strongly depends on the traits between fermentation and respiration. We use *S. pombe* as a model system, as others, like the Covo lab, use laboratory (non-industrial) strains of *S. cerevisiae*. Fission yeast is only remotely related to budding yeast (which is a potent model for universal metabolic principles), and shows features that provide valuable complementary insights, especially into mitochondrial metabolism. Mitochondria of fission yeast form a dynamic network along microtubules which mediate their inheritance, as is the case in multicellular eukaryotes [6]. The fission yeast mitochondrial genome is compact (~20 kb, 11 protein-coding genes) and mitochondrial RNA processing is similar to animal cells [7]. Fission yeast can grow using either respiration or fermentation but, in contrast to budding yeast, does not thrive in strictly anaerobic conditions [8]. The basic science learned with these non-industrial yeast strains has so far proven pivotal far beyond the respective research communities.

Yeast research has frequently informed the medical community; for example, the Warburg effect in cancer cells is a balancing problem between respiration and aerobic glycolysis. Cellular aspects of mitochondrial metabolism are typically highly conserved. We agree with Reviewer #3 who states that there is wider significance in highlighting the crosstalk between glycolysis and the respiratory chain under conditions of aerobic glycolysis. Our work addresses an important, but under-investigated, area of cellular physiology. Mitochondrial dysfunctions play roles in a range of age-related diseases. Because of the central role of mitochondria in cellular metabolism, mitochondrial dysfunction produces pleiotropic molecular and organism-level phenotypes that can be difficult to explain. That is why it is important to trace molecular consequences of a respiratory block. We analyse effects of mitochondrial dysfunction on arginine metabolism which is difficult to study in multi-cellular models where arginine is an essential media addition. However, as pointed out in the Conclusion, these

processes may be conserved, and our results could spark scientists working with human subjects to consider it.

Reviewer #2 (Evidence, reproducibility and clarity (Required)):

The studies in this manuscript address the question as to why fission yeast partially rely on respiration for rapid proliferation even in the presence of a fermentable carbon source. The study addresses this question with a series of experiments which meticulously delineates the factors. In the first study, it is shown that supplementation of the growth medium with arginine alone is sufficient to relieve the need for respiration for rapid proliferation. They then determined, using the collection of yeast null deletion mutants, that loss of function of a number of genes mimics effect by the loss of respiration and the effect of arginine. Many of these genes were in the arginine pathway, as expected, and many were in the respiratory chain or in those related to the ATP synthase. In the last study, the authors investigate gene expression profiles in response to a block in respiration in the presence and absence of arginine. These data indicate that there is also an effect on the genes involved in regulation of TOR. The principal conclusion of these studies is that in fission yeast, the respiratory chain is needed for rapid cell growth due to the role of the TCA in amino acid synthesis - specifically, that of Arg and its metabolic products.

*Overall, this is a well controlled study that makes sound conclusions. The main issues I have are related to Fig. S3 where genes are identified that when deleted, make the cells respond to Arg supplementation. In some cases, the discrimination factor is small where just a small effect is seen- as with *shy1* and *rrf1*. In other cases, the effects are not equivalent despite the fact that they are related to the same pathway or enzyme. For instance, the effect by loss of *atp1* is much smaller than that of *atp12* or *atp7* and *atp2* is not even listed. While this is not significant enough to prevent from publication, it does suggest that something more is happening.*

Reply: We thank the Reviewer for the positive overall assessment. For Figure S3, we used 44 mutants identified in our previous study as affecting growth on a non-fermentable carbon source in both prototroph and auxotroph strain backgrounds [9]. The purpose of this figure is to confirm the link between respiration and arginine levels using independent deletion mutants, selected by criteria other than arginine auxotrophy. The 44 mutants have been identified in a high-throughput screen and *atp2* did not pass the threshold applied [9], which is why it is not used here. As mentioned in the manuscript, some of the mutants do not exhibit a phenotype in liquid EMM medium, but they have all been identified as affecting growth on non-fermentable carbon sources. Please note also that most of these mutants have not been identified in the arginine supplementation screen using our threshold. Thus, we do agree that the relationship between respiratory ability and arginine synthesis is more complex and will need further investigation. However, blocking the electron transport chain with antimycin A as well as many mutations that affect respiratory growth, including those that affect the mitochondrial ATPase, do result in arginine deficiency. We hypothesise that the effect of a given mutant on the mitochondrial membrane potential or redox homeostasis translates to the arginine level. We now propose this possibility in the Discussion.

We also point out that all strains used have been selected from the *S. pombe* deletion library or from the library backcrossed with a wild-type strain to obtain a prototroph library [9]. These libraries might accumulate secondary mutations that could change phenotypes, as has been shown in budding yeast. Moreover, all high-throughput screens unavoidably produce some false positive and false negative results, so we draw our conclusions based on significant overrepresentations of groups of mutants rather than on the evaluation of individual mutants. We are certain, however, that our results support the conclusion that respiration and amino-acid synthesis are connected and that amino-acid synthesis limits fully fermentative metabolism.

Also, why are mutation in genes encoding the ATP synthase related to the electron transport chain?

Reply: We do not really understand this comment. We looked in the manuscript but did not find any statement that claims this relationship between the ATP synthase and ETC.

A minor correction - I think - is on p.7, last para, "... although arginine is essential in humans..." should be "... although arginine is not essential in humans..."

Reply: We corrected this sentence into: "Arginine is semi-essential in humans, and is an essential supplement for mammalian cell cultures, rendering it difficult to study the impact of respiration on arginine levels in this system."

Reviewer #2 (Significance (Required)):

The findings in this study are significant if only related to yeast metabolism, but highly significant if it can be related to mammalian metabolism. There are a number of human conditions where mitochondrial function is important and the loss there-of is detrimental. On the surface, this seems to be an obvious statement, but it is much less clear. For instance, many cancer cells grow without usage of the respiration - referred to as the Warburg effect. Thus, a priori, cells do not need the respiratory chain for ATP synthesis. There is loss of mitochondrial function with aging and in many neurodegenerative diseases. The studies here provide a possible link for the need of respiratory chain with cell growth; it is unclear how this will be related to human cells.

Outside the references listed in the discussion of this manuscript, there is some relationship of this work with the studies on the genetics and biochemistry of "petite negative" yeasts. These studies looked to address why some yeast cannot lose their mitochondrial DNA while "petite positive" yeast can lose their mitochondrial DNA. The study here is the first study to provide a possible link to understanding this dichotomy.

The audience that would be interested in this work are likely to be those that study yeast mitochondria metabolism and genetics. However, if this study furthered and related to human cells or diseases, then this can be a high impact paper.

Reply: We agree and propose that principles reported here may be more widely conserved and broadly relevant. Only time will tell, of course. See also our reply to the Significance section of Reviewer #1.

My expertise is more in yeast genetics, mitochondria, ATP synthesis, and protein structure. I can't say that I am expert on any specific aspect in this report, but I am confident that I understand the rationale, methods, and results.

Reviewer #3 (Evidence, reproducibility and clarity (Required)):

This is an interesting study that describes the metabolic contribution of the Krebs cycle to fission yeast growth under fermentative conditions (i.e. with glucose as carbon source). The authors build on their observation that chemical interference with the electron transport chain with antimycin has strong effects on cell growth in glucose minimal medium but much less effect in rich medium with glucose as carbon source. The authors then convincingly show that this effect is a specific result of an arginine deficit in the challenged cells, and a genome-wide screen reinforces the conclusion by showing a strong correlation between arginine 'auxotrophy' and defects in mitochondrial functions. The primary conclusion of the work is that fission yeast respiration compensates for arginine deficits resulting from fermentative growth, that this compensation translates to alleviation of deficits in arginine-derived amino acids as well (primarily glutamine, lysine and glutamate), and that these deficits are correlated with but independent of TOR regulation.

In sum, this is a nicely comprehensive and systematically constructed body of work, and it is logically consistent across the multiple independent lines of investigation that were followed. The unexpected but attractive finding that further elevates the impact of this MS is that the proteins induced by arginine deficit (as compensatory mechanism) are underrepresented in arginine, lysine, glutamine and glutamate.

Overall, a well-done analysis that addresses an important, but under-investigated, area of cellular physiology.

Reply: We thank the Reviewer for this very positive assessment.

Reviewer #3 (Significance (Required)):

The major significance of this work is that it addresses an important, but under-investigated, area of cellular physiology. That is, the crosstalk between glycolysis and the respiratory chain under conditions of aerobic glycolysis. The demonstration that this crosstalk focuses down on the specific target of arginine availability, and consequently to the availability of amino acids derived from arginine (lysine, glutamine, glutamate) is interesting. Of particular interest is the fact that, while these amino acids activate TOR, and the TOR pathway is inhibited by the respiration block in aerobically fermenting cells, that relationship is correlative yet independent. That the gene products induced to compensate for these amino acid deficits are themselves underrepresented. This general question is of significant contemporary interest and it will be of interest to a broad audience that will include researchers in the biochemistry, cell cycle and cancer cell biology fields. The manuscript is clearly written and the conclusions well-supported. I pose three issues for consideration by the authors:

Reply: Thank you, we agree with the wider significance of these findings beyond fission yeast.

(i) The manuscript would be strengthened by a flux experiment where mass-tagged arginine is fed to antimycin-treated cells and the rate of its conversion to lysine, glutamine and glutamic acid is followed in a time course. Those data could then be correlated to the existing transcriptomic data. This experiment would provide a direct measurement of the kinetics of conversion which is otherwise inferred from the existing data. Since the readout used is an LC-MS measurement, this should be a straightforward experiment.

Reply: We agree in principle that this would be a meaningful complementary experiment, although not that straightforward. Currently we are not able to perform wet-lab experiments. Please note that the rates of conversion of supplementary arginine to other amino acids, albeit not with antimycin A treatment, have been measured by Bicho et al. [10]. They show that conversion in this condition is highly efficient in fission yeast. It is likely that the rate of arginine conversion in antimycin-treated cells will be similar.

(ii) What happens to the arginine auxotrophy in antimycin-treated cells if the natural pyruvate oxidase mutation in fission yeast is corrected? Presumably, this correction would rescue the arginine auxotrophy.

Reply: For several reasons, we think that in the *pyk1* allele-replacement strain, we would observe arginine auxotrophy as well. The natural variant found in the standard laboratory strain limits the maximum possible flux from pep to pyruvate, while still being able to support healthy, albeit slightly slower, growth. The effect of the mutation is the strongest when flux is highest, i.e. in rich glucose media with yeast extract [11]. Specifically, in those conditions the standard lab strain becomes limited by pyruvate-kinase flux, grows slower and relies more on respiration to meet energy demands. Moreover, the effect of the *pyk1* mutation is much weaker on minimal media (with ammonium as sole nitrogen source) and absent on carbon sources that support lower growth rates (Fig. 5C in [11]). Thus, there are no strong physiological differences between the laboratory strain and the *pyk1* allele-replacement strain in minimal media: the laboratory strain still exhibits mainly fermentative metabolism and produces ethanol (see [12] for example). The laboratory strain, even though it has lower pyruvate-kinase flux, still produces excess pyruvate which it needs to get rid of, so clearly pyruvate is not the limiting factor for arginine production.

In the current manuscript, we show that the factor limiting growth under respiration deficiency in minimal media is not energy production but arginine biosynthesis. So we do not expect that using a strain with higher Pyk1 activity will impact arginine auxotrophy. Although we are not able to perform additional experiments at this time, our unpublished data indicate that growth of the *pyk1* allele-replacement strain is indeed affected by antimycinA in minimal media similarly to the laboratory strain (see **Figure2**).

Figure 2. Growth of both the laboratory strain (Wt) and the *pyk1* allele-replacement strain (*pyk_A h-*) is inhibited by antimycin A in minimal medium.

(iii) Why did the *mss1* mutant fail to recapitulate the amino acid deficiencies seen in antimycin-treated cells and in the four other respiratory chain mutants analyzed?

Reply: The *mss1* deletion actually recapitulates the phenotype, just to a lesser extent, because the growth effect after adding arginine is smaller compared to other strains used. We do indeed observe small decreases in cellular levels of arginine and ornithine in the *mss1* deletion.

References cited:

1. Druseikis M, Ben-Ari J, Covo S (2019) The Goldilocks effect of respiration on canavanine tolerance in *Saccharomyces cerevisiae*. *Curr Genet* **65**: 1199–1215.
2. Liu Z, Butow RA (1999) A Transcriptional Switch in the Expression of Yeast Tricarboxylic Acid Cycle Genes in Response to a Reduction or Loss of Respiratory Function. *Mol Cell Biol* **19**: 6720–6728.
3. Mülleder M, Calvani E, Alam MT, Wang RK, Eckerstorfer F, Zelezniak A, Ralser M (2016) Functional Metabolomics Describes the Yeast Biosynthetic Regulome. *Cell* **167**: 553-565.e12.
4. Chiron S, Gaisne M, Guillou E, Belenguer P, Clark-Walker GD, Bonnefoy N (2007) Studying mitochondria in an attractive model: *Schizosaccharomyces pombe*. *Methods Mol Biol* **372**: 91–105.
5. Chen XJ, Clark-Walker GD (1999) The petite mutation in yeasts: 50 years on. *Int Rev Cytol* **194**: 197–238.
6. Yaffe MP, Stuurman N, Vale RD (2003) Mitochondrial positioning in fission yeast is driven by association with dynamic microtubules and mitotic spindle poles. *Proc Natl Acad Sci U S A* **100**: 11424–11428.
7. Schäfer B, Hansen M, Lang BF (2005) Transcription and RNA-processing in fission yeast mitochondria. *RNA* **11**: 785–795.
8. Heslot H, Goffeau A, Louis C (1970) Respiratory metabolism of a ‘petite negative’ yeast *Schizosaccharomyces pombe* 972h-. *J Bacteriol* **104**: 473–481.
9. Malecki M, Bähler J (2016) Identifying genes required for respiratory growth of fission yeast. *Wellcome Open Res* **1**: 12.
10. Bicho CC, Alves FDL, Chen Z a, Rappsilber J, Sawin KE (2010) A genetic engineering solution to the ‘arginine conversion problem’ in stable isotope labeling by amino acids in cell culture (SILAC). *Mol Cell Proteomics* **9**: 1567–1577.
11. Kamrad S, Grossbach J, Rodríguez-López M, Mülleder M, Townsend S, Cappelletti V, Stojanovski G, Correia-Melo C, Picotti P, Beyer A, et al. (2020) Pyruvate kinase variant of fission yeast tunes carbon metabolism, cell regulation, growth and stress resistance. *Mol Syst Biol* **16**: 770768.
12. Ozaki A, Konishi R, Otomo C, Kishida M, Takayama S, Matsumoto T, Tanaka T, Kondo A (2017) Metabolic engineering of *Schizosaccharomyces pombe* via CRISPR-Cas9 genome editing for lactic acid production from glucose and cellobiose. *Metab Eng Commun* **5**: 60–67.

Dear Jürg,

Thank you for submitting your revised manuscript, which was previously reviewed at Review Commons. I have now heard back from one of the original referees. The referee finds that the manuscript was significantly improved after revision and recommends publication. Before I can accept the manuscript, I need you to address some minor points below:

- Please add 'Author Contributions' and 'Conflict of Interest' sections.
- Please fill out and include an author checklist as listed in our online guidelines (<https://www.embopress.org/page/journal/14693178/authorguide>)
- Please include a Data Availability Section - if it is not applicable, make a statement that no data were deposited in a public database. Primary datasets (and computer code, where appropriate) produced in this study need to be deposited in an appropriate public database (see). Please make sure that the data are accessible.
- Please convert the reference style to Harvard style. Please see <https://www.embopress.org/page/journal/14693178/authorguide#referencesformat>
- We noted that the UK MRC and The Wellcome Trust grants both have the same number. Please check.
- Please upload figures as separate files - one file per figure. Please provide the figures in higher resolution.
- We noted that Figs S4 panels and Fig S5A are currently not called out in the text.
- Please clarify in the figure legend that the same graphs were presented in Figures 2C and S3.
- Please change the 5 supplemental figures into EV Figures and remember to update the figure callouts accordingly.
- There are 5 supplemental tables. Tables S1-S4 should be Dataset files, and Table S5 should be Table EV1.
- Papers published in EMBO Reports include a 'Synopsis' to further enhance discoverability. Synopses are displayed on the html version of the paper and are freely accessible to all readers. The synopsis includes a short standfirst summarizing the study in 1 or 2 sentences that summarize the key findings of the paper and are provided by the authors and streamlined by the handling editor. I would therefore ask you to include your synopsis blurb.
- In addition, please provide an image for the synopsis. This image should provide a rapid overview of the question addressed in the study but still needs to be kept fairly modest since the image size cannot exceed 550x400 pixels.
- Our production/data editors have asked you to clarify several points in the figure legends (see attached document). Please incorporate these changes in the attached word document and return it with track changes activated. Please upload the file in word format as manuscript text.

Thank you again for giving us to consider your manuscript for EMBO Reports, I look forward to your minor revision.

Kind regards,

Deniz

--

Deniz Senyilmaz Tiebe, PhD
Editor

EMBO Reports

Referee #3:

i have looked over the revised MS and the rebuttal letter. i am satisfied that this MS is now suitable for publication in EMBOR. I disagree with the criticisms regarding diminished novelty and limited impact. it is both novel and impactful. no further revisions necessary.

Rev_Com_number: RC-2020-00189

New_manu_number: EMBOR-2020-50845V1

Corr_author: Bähler

Title: Mitochondrial respiration provides amino acids during fermentative proliferation in fission yeast

Dear Deniz,

Thank you for your decision. We have addressed all the issues you raised as pointed out below.

- Please add 'Author Contributions' and 'Conflict of Interest' sections. - **done**
- Please fill out and include an author checklist as listed in our online guidelines - (<https://www.embopress.org/page/journal/14693178/authorguide>) – **done**
- Please include a Data Availability Section - if it is not applicable, make a statement that no data were deposited in a public database. Primary datasets (and computer code, where appropriate) produced in this study need to be deposited in an appropriate public database (see <http://embor.embopress.org/authorguide#dataavailability>). Please make sure that the data are accessible. - **done**
- Please convert the reference style to Harvard style. Please see <https://www.embopress.org/page/journal/14693178/authorguide#referencesformat> - **done**
- We noted that the UK MRC and The Wellcome Trust grants both have the same number. Please check. - **It is correct to have the same grant numbers here; this is a peculiarity of the Francis Crick Institute.**
- Please upload figures as separate files - one file per figure. Please provide the figures in higher resolution. - **done**
- We noted that Figs S4 panels and Fig S5A are currently not called out in the text. - **corrected**
- Please clarify in the figure legend that the same graphs were presented in Figures 2C and S3. - **done**
- Please change the 5 supplemental figures into EV Figures and remember to update the figure callouts accordingly. – expanded view - **done**
- There are 5 supplemental tables. Tables S1-S4 should be Dataset files, and Table S5 should be Table EV1. **Tables S1-S4 have been divided to four Dataset files. Table S4 is now Table EV2, and the table embedded previously in figure S2 (now Fig EV2) is now Table EV1.**
- Papers published in EMBO Reports include a 'Synopsis' to further enhance discoverability. Synopses are displayed on the html version of the paper and are freely accessible to all readers. The synopsis includes a short standfirst summarizing the study in 1 or 2 sentences that summarize the key findings of the paper and are provided by the authors and streamlined by the handling editor. I would therefore ask you to include your synopsis blurb. – **Synopsis included in manuscript text (before Abstract)**
- In addition, please provide an image for the synopsis. This image should provide a rapid overview of the question addressed in the study but still needs to be kept fairly modest since the image size cannot exceed 550x400 pixels. – **done**
- Our production/data editors have asked you to clarify several points in the figure legends (see attached document). Please incorporate these changes in the attached word document and return it with track changes activated. Please upload the file in word format as manuscript text. – **these queries are answered in the manuscript file**

Best wishes,

-Jürg

Dear Jürg,

Thank you for submitting your revised manuscript. I have now looked at everything and all is fine. Therefore I am very pleased to accept your manuscript for publication in EMBO Reports.

Congratulations on a nice study!

Kind regards,

Deniz

--

Deniz Senyilmaz Tiebe, PhD
Editor
EMBO Reports

You will receive proofs by e-mail approximately 2-3 weeks after all relevant files have been sent to our Production Office; you should return your corrections within 2 days of receiving the proofs. Please inform us if there is likely to be any difficulty in reaching you at the above address at that time. Failure to meet our deadlines may result in a delay of publication, or publication without your corrections. All further communications concerning your paper should quote reference number EMBOR-2020-50845V2 and be addressed to emboreports@wiley.com.

If sequence, structural, or microarray data is included, the data must be deposited with the appropriate databases. Please follow our instructions to authors, which can be found at: http://embor.msubmit.net/html/embor_author_instructions.html.

- ORCID

EMBO Press encourages all authors and reviewers to associate an Open Researcher and Contributor Identifier (ORCID) to their account. ORCID is a community-based initiative that provides an open, non-proprietary and transparent registry of unique identifiers to help disambiguate research contributions.

Rev_Com_number: N/a

New_manu_number: EMBOR-2020-50845V2

Corr_author: Bähler

Title: Mitochondrial respiration provides amino acids during fermentative proliferation in fission yeast

Corresponding Author Name: Jurg Bahler; Michał Malecki

Manuscript Number: EMBOR-2020-50845V1 | [RC-2020-00189]